# Association between Knowledge about Anemia, Food Consumption Behaviors, and Hematocrit Level among School-Age Children in Nakhon Si Thammarat Province, Thailand

**Pastraporn Kaewpawong** [1], **Kiatkamjorn Kusol** [1,*], **Onuma Bunkarn** [2] and **Sirikran Sutthisompohn** [1]

1   School of Nursing, and the Excellence Center of Community Health Promotion, Walailak University, 222 Thaiburi, Thasala District, Nakhon Si Thammarat 80161, Thailand
2   Saikhueng Sub-District Health Promoting Hospital, Suratthani 84000, Thailand
*   Correspondence: kkiatgum@gmail.com

**Correction Statement:** This article has been republished with a minor change. The change does not affect the scientific content of the article and further details are available within the backmatter of the website version of this article.

**Abstract:** Anemia is a significant public health problem among children, especially school-age children, because their body quickly produces red blood cells to provide sufficient blood volume with plasma expansion to maintain blood concentration. This research aimed to study the anemia situation, knowledge about anemia, food consumption behaviors, and the association between knowledge, food consumption behaviors, and hematocrit level among school-age children in primary school, in Thasala district, Nakhon Si Thammarat Province. This is a descriptive study among 408 students in grades 4 to 6, aged 9–12 years. Research instruments included the children's demographic data of the children, their knowledge about anemia, and food consumption behaviors, and hematocrit assessment. Data were analyzed using descriptive, Spearman's rank correlation coefficient, and logistic regression statistics. The results revealed that 23.2% of the samples had anemia, 0.98% had moderate anemia, and 22.22% had mild anemia. The children knew about anemia, with the mean score at a moderate level (mean = 6.63; SD = 2.51) out of 10. The mean score on food consumption behaviors was at a moderate level (mean = 17.49; SD = 3.68) out of 24. There were significantly positive correlations between the knowledge about anemia and hematocrit level at a moderate level (r = 0.45, $p < 0.001$). Food consumption behaviors were moderately correlated with hematocrit level (r = 0.40, $p < 0.001$). Confirmation with logistic regression found that knowledge about anemia (OR = 9.15, 95% CI: 4.57–18.34) and food consumption behaviors (OR = 19.09, 95% CI: 9.71–37.53) were significantly associated with hematocrit level. Conclusions: This study showed that knowledge about anemia and appropriate food consumption behaviors are associated with hematocrit levels. Enhancing knowledge about anemia and food consumption behaviors may reduce the prevalence of anemia in school-age children. Health care providers in primary care should provide health education and encourage children to eat sufficient food.

**Keywords:** knowledge; food consumption behaviors; hematocrit level; school-age children

## 1. Introduction

Anemia in children is a significant public health problem worldwide in developing and developed countries. However, the prevalence is exceptionally high in developing countries. A World Health Organization (WHO) survey found that 30 percent of the population, or more than 1.3 billion people, were anemic [1]. Approximately 500–600 million people, or one-third of the world's population, are affected by anemia and develop iron deficiency anemia but may be asymptomatic because the body still has iron stored in reserve [2,3]. Still, chronic anemia, especially in children, is one of the causes of illness and risk of death because it lowers immunity and increases susceptibility to infection, more so than in adults [4]. In Latin America and the Caribbean, the prevalence of anemia among school-age

children was 17.49% [5]. A study in southwestern Ethiopia found that 37.6% of school-age children had anemia, while 18.1% had mild anemia, and 19.5% had moderate anemia [6]. In Thailand, the prevalence of anemia in school-age children was 27% based on the WHO-prescribed red blood cell concentration criteria (Hb < 12 g/dL and Hct < 36%) [1]. In particular, a survey by the Division of Nutrition, Department of Health found that the prevalence of anemia among 6-year-old children was 31.1%. This was greater than the acceptable limit of 10%, based on the Thailand Department of Health criteria [7].

Anemia causes a decrease in growth, cognitive processes, intelligence, and learning efficiency among school-age children. Anemia is one of the top five factors affecting the health of Thai children aged 0–14 years. A study on the impact of chronic anemia among school-age children found that these children exhibited slower physical growth, stunting at the growth thresholds, and developmental delays compared to others in the same age group [8,9]. Additionally, anemia affects the immune system, causing reduced defense against pathogens and reduced physical activity and leading to infection, making children constantly sick [10]. There is also a manifestation of symptoms including tiredness, fatigue, a decreased ability to exercise, and a decreased capacity to learn [11]. Long-term effects can include neurodevelopmental disorders [12]. In addition, emotional and behavioral effects in children included being easily irritable, fearful, and startled. They may also have a lack of confidence and motivation [13].

One factor causing school-age children to suffer from anemia is a decrease in red blood cell production [14]. In developing countries, food consumption behaviors of growing school-age children are insufficient to meet the body's needs. In addition, food consumption behaviors include avoiding diets containing green leafy vegetables or meat and habitual consumption of crispy snacks and soft drinks. Other factors include tooth decay, chronic blood loss from hookworm infections, lice, and stomach ulcers. Anemia is more common among mothers and low-income families, and those with lower parental education levels are more susceptible than those with higher incomes and education [6,11,15,16].

Control and prevention of anemia rely on both medication and non-medication formulations. School-age children, 6–14 years, should take 60 mg iron tablets once a week [17]. The Department of Health, Ministry of Public Health, has issued a policy for the school health sector to focus on nutrition education and screening for anemia. However, it is still impossible to cover all areas due to a low budget and lack of clarity in implementing the policy. As a result, screening and prevention of anemia in school-age children are not comprehensive in many areas [18,19]. In addition, school-age children are paler than the standard set by the Department of Health, Thailand. Studies among preschool children have investigated the effectiveness of nutrition promotion programs on their growth in child development centers. As a result of implementing the programs, preschool children grew better. Their asymmetry reached higher standards. Parents and teachers increased nutrition promotion attitudes and practices [20]. For one group of high school students, nutrition promotion programs affected their nutritional behaviors and red blood cell concentrations, leading to improved food consumption behavior and increased concentrations [21,22]. In school-age children, there was a study on the dietary behaviors following a modification program among children with overnutrition, which found that the children understood food consumption and improved their food consumption behaviors [23,24]. Furthermore, a study on the Phasi Charoen people found that most consumers had a moderate level of expertise regarding food consumption behaviors.

The most common food consumption behavior among school-age children was drinking sweetened milk. The media has a considerable influence on the food consumption behaviors of children. School-age children suffering from overweight and obesity had inappropriate food intake [25].

Because there are relatively few studies on anemia among school-age children, most studies focus on early childhood and adolescence. It is important to understand the knowledge and food consumption behaviors that can protect against anemia in children [26]. According to commercial media, school-age children still lack an understanding of food

consumption behaviors, although this group can learn many things regarding their health and the environment around them. In this study, a hematocrit test for assessing anemia and assessing knowledge and food consumption behaviors were investigated. The researchers noted the necessary information regarding the anemia situation among school-aged children and hope that this study can accurately reflect the anemia situation, knowledge, and food consumption behaviors of school-age children and their hematocrit levels. This research aimed to study the anemia situation, knowledge about anemia, food consumption behaviors, and the association between knowledge, food consumption behaviors, and hematocrit level among school-age children in primary school in the Thasala district. The results of this study will help teachers and primary health care providers to design and implement health education programs for students, following the public health policy, to decrease anemia prevalence within the acceptable limit of 10%.

## 2. Materials and Methods

### 2.1. Study Design

This study was cross-sectional descriptive research. The research population included school-age students in grades 4 to 6, aged 9 to 12 years, in Nakhon Si Thammarat Province. The researchers were randomized and assigned to the Thasala district. The Thasala district's child population totals 3691 studying in primary schools. The researchers calculated the sample size using the Krejcie and Morgan formula and added about 15% to collect comprehensive data, generating 408 research samples.

### 2.2. Measures

Recruitment and sampling: Stratified sampling was randomly selected, followed by a random sample of schools from 10 subdistricts. Simple random selection was used to determine the 5 schools to be studied in selected 3 subdistricts to meet the predetermined size of a sample group and represent each community, followed by simple random sampling in compliance with the inclusion criteria, yielding a group of 408 research samples. The sample was school-age children in grade 4 = 145, grade 5 = 139, and grade 6 = 124.

The inclusion criteria included no congenital diseases, no blood diseases, no experience of operations or blood transfusion, and no accidents inflicting blood loss over the past three months. Furthermore, in girls who have not reached menstrual age, participants also need to be able to read, write and verbally communicate. Both the children and their parents consented to participate in the study, which preceded the process entailing the documentation of the children's history, the hematocrit test, the assessment of their knowledge about anemia and its prevention, and their food consumption behaviors.

### 2.3. Research Instruments

1. Demographic data questionnaire: This was employed to collect demographic data comprising sex, age, weight, height, educational levels, number of siblings of the school-age children, religion, family roles, and occupations of the caregivers.

2. Assessment of anemia: The result of the hematocrit level from a packed red cell volume (PCV) was assessed as normal when Hct $\geq$ 36% and anemic when Hct < 36% (mild = 30–35%, moderate = 21–29%, severe = <21%) [6,7].

3. Knowledge assessment of anemia and its prevention: The researchers adapted the original knowledge assessment version by Chanjira Saengngern, which had a reliability score 0.89 for precision and consisted of 10 items (each of which was to be answered with Yes or No), yielding a total score of 10 points [27]. The scores can be divided into three levels. One correct answer generated one point, while an incorrect answer was given zero points. Consequently, scores lower than 60% (<6 points) demonstrated a low level of knowledge, 60 to 79% (6–7.9 points) showed a moderate level, and 80% or higher (8–10 points) indicated a good level of knowledge [28].

4. Assessment of food consumption behaviors in children: The researchers adapted the assessment from the Department of Health, Bureau of Nutrition, Ministry of Public

Health, consisting of 6 items and yielding a total score of 24 points, using a 4-point rating detailed as follows [18].

| | |
|---|---|
| Always engage in that behavior, averaging 5 to 7 times or 5 to 7 days a week | =4 |
| Occasionally engage in that behavior, averaging 3 to 4 times or 3 to 4 days a week | =3 |
| Rarely engage in that behavior, averaging 1 to 2 times or 1 to 2 days a week | =2 |
| No engagement in that behavior in one week | =1 |

The 24 points were divided into three levels, detailed as follows:

| | |
|---|---|
| Score 18–24 | =Good |
| Score 12–17 | =Moderate |
| Score 6–11 | =Low |

For the reliability measurement, the assessment of the knowledge about anemia and food consumption behaviors was later distributed to 30 other school-age children sharing the same characteristics as the research samples. The yielded scores for reliability were 0.77 and 0.73, respectively.

*2.4. Ethical Considerations*

The researchers conducted the study following the Declaration of Helsinki. All procedures performed in this study involving human participants followed the ethical standards of the Ethical Institutional Consideration. This study received approval from the Ethics Committee on Human Research at Walailak University on 30 April 2020 (WUEC-20-096-01) as required by the process before data collection. Informed consent was obtained by the researchers from all individual participants included in the study.

*2.5. Statistical Analysis*

This study analyzed the statistics using SPSS software (Version 24) for Windows™ (IBM Corporation, New York, NY, USA). The statistics employed the following:

1. Descriptive statistics were used to analyze the demographic data, anemia situation, knowledge about anemia, and food consumption behaviors entailing frequencies, percentages, means, and standard deviations (S.D.).

2. The association between knowledge, food consumption behaviors, and hematocrit level was analyzed using the Kolmogorov–Smirnov test, demonstrating a difference from a normal distribution. The association was analyzed using Spearman's rank correlation coefficient and using binary logistic regression statistics, determining statistical significance at 0.05.

**3. Results**

The results of the descriptive research carried out to study the demographic data, the anemia situation, knowledge about anemia, and food consumption behaviors and the associations between knowledge about anemia, food consumption behaviors, and hematocrit level in school-age children are as follows:

The demographic data of the school-age children in grades 4–6 indicated that 53.9 percent of the children were females and 46.1 percent were males. Sixty percent were aged 11–12 years, followed by an age of 9–10 years accounting for forty percent. For religion, 56.4 percent were Muslim, followed by Buddhists, accounting for 43.6 percent. The height index for age and weight index for age reveal that the school-age children achieved the age-appropriate standard in 70.3%, and 69.4% of cases, respectively. Caregivers included parents, accounting for 53.2 percent, followed by grandparents at 19.4 percent. Regarding occupations of the caregivers, 45.6 percent were wage earners, followed by trade at 20.1 percent. Finally, 30.4 percent of the children had more than three siblings, followed by 27.6 percent having two siblings.

The 408 school-age children demonstrated a hematocrit level range of 25–47%, mean = 37.66%, and SD = 3.35, with 76.8 percent not having anemia (normal) and 23.2 percent having anemia. Those experiencing anemia had a hematocrit (Hct) value falling in the range of 25–35 %, with a mean Hct of 33.07%. Based on the anemia condition

levels, a total of 95 school-age children had anemia, 91 (22.22%) of whom had a mild level of anemia, and 4 (0.98%) of whom had a moderate level, as shown in (Table 1).

**Table 1.** Number and percentage of the samples with hematocrit levels of normal and anemic (n = 408).

| Hematocrit Level | Normal n (%) | Anemia n (%) | Total n (%) |
|---|---|---|---|
| School-age grade 4 | 100 (24.6) | 45 (11.0) | 145 (35.6) |
| School-age grade 5 | 107 (26.2) | 32 (7.8) | 139 (34.0) |
| School-age grade 6 | 106 (26.0) | 18 (4.4) | 124 (30.4) |
| Overall<br>Hct (Range = 25–47, mean = 37.66, S.D. = 3.35) | 313 (76.8) | 95 (23.2)<br>Hct (Range = 25–35, Mean = 33.07, S.D. = 2.74)<br>Mild = 91 (22.22%)<br>Moderate = 4 (0.98%) | 408 (100) |

The mean score of the knowledge assessment about anemia in school-age children was 6.63 (SD = 2.51) out of 10, indicating a moderate level. The mean score of the food consumption behaviors of the school-age children was 17.49 (SD = 3.68) out of 24, showing a moderate level, as demonstrated in (Table 2).

**Table 2.** Mean score and standard deviation (SD) of the knowledge about anemia and food consumption behaviors of the samples (n = 408).

| Variable | Min | Max | Mean | S.D. | Level |
|---|---|---|---|---|---|
| Knowledge about anemia | 1 | 10 | 6.63 | 2.51 | Moderate |
| Food consumption behaviors | 8 | 24 | 17.49 | 3.68 | Moderate |

The relationship between the knowledge about anemia, food consumption behaviors, and anemia prevention was assessed using Spearman's rank correlation coefficient statistics. This study found that knowledge about anemia and food consumption behaviors were associated with hematocrit levels at 0.45 and 0.40 ($p < 0.001$), respectively, at a moderate level, as shown in (Table 3).

**Table 3.** Relationships between knowledge about anemia, food consumption behaviors, and hematocrit level of the samples (n = 408).

| Variable | r | *p*-Value | Level |
|---|---|---|---|
| Knowledge about anemia | 0.45 | 0.001 *** | Moderate |
| Food consumption behaviors | 0.40 | 0.001 *** | Moderate |

*** $p < 0.001$.

Furthermore, using binary logistic regression analysis suggested that children having a knowledge score of more than 6 in a moderate to good level was significantly associated with hematocrit level (OR = 9.15 (95% CI: 4.57–18.37)). In terms of food consumption behaviors, the children exhibited a good level significantly associated with hematocrit level (OR = 19.09; 95% CI: 9.71–37.53), as displayed in Table 4.

**Table 4.** Binary logistic regression analysis for exploring knowledge and food consumption behaviors factors associated with hematocrit level (n = 408).

| Variable | | Hematocrit Level Normal N (%) | Hematocrit Level Anemia N (%) | B | SE | Wald | df | Sig | EXP(B) | 95% CI |
|---|---|---|---|---|---|---|---|---|---|---|
| Knowledge of anemia | Score < 6 | 95 (54) | 81 (46) | | | | | | 1 | 1 |
| | Score > 6 | 218 (94) | 14 (6) | 2.21 | 0.36 | 38.95 | 1 | 0.00 ** | 9.15 | 4.57–18.37 |
| Food consumption behaviors | Score < 18 | 58 (41.7) | 81 (58.3) | | | | | | 1 | 1 |
| | Score > 18 | 255 (94.8) | 14 (5.2) | 2.95 | 0.35 | 73.06 | 1 | 0.00 ** | 19.09 | 9.71–37.53 |

Cox and Snell R Square = 0.37, Nagelkerke R Square = 0.56, ** $p < 0.001$.

## 4. Discussion

The study results regarding the anemia situation, knowledge about anemia, food consumption behaviors, and hematocrit level among school-age children in the Thasala district, Nakhon Si Thammarat, suggested that 23.2 percent had anemia, most of whom had a mild level. The knowledge assessment about anemia and food consumption behaviors for anemia prevention demonstrated that the target group was at a moderate level for both topics. Furthermore, the study results showed that knowledge about anemia and appropriate food consumption behaviors were associated with hematocrit level in school-age children.

The prevalence of anemia among primary school children in Nakhon Si Thammarat was 23.2 percent, indicating a mild to moderate level. Nevertheless, without proper screening, children would not be aware of their anemia since no apparent symptoms manifested due to the mild to average levels of anemia resulting from the body's iron stored in reserve [1]. The prevalence of anemia among school-age children revealed in this study seemed to be higher than the prevalence investigated in the conditions among Asian adolescents in Indonesia, the Republic of China, and Kuwait, where the anemia prevalence was 14, 12, and 8.06 percent, respectively [29–31]. However, it was lower than that found in low-income countries such as Ethiopia, Ghana, and Nepal, where the anemia prevalence was 37.3, 29.4, and 31 percent, respectively [32–34]. Anemia is still a common problem in developing countries. Social, cultural, and economic disparity is undeniably associated with accessibility to iron-rich foods and public health systems. Compared to the national level, it was found that Thailand's school-age children's anemia conditions exceeded the acceptable level of 10 percent as determined by the Department of Health. This indicated that anemia among the target group of this study was two times higher than the standard set [35]. Currently, anemia still exists in developing countries and needs to be addressed with a collaborative effort from multiple parties including schools, families, the children themselves, and health professionals. Treatment for this condition has to be non-medication-reliant. According to the Bureau of Nutrition's policy, iron and folic acid supplements should be incorporated once a week [35].

Regarding knowledge about anemia and food consumption behaviors for anemia prevention, the sample group exhibited knowledge about anemia and food consumption behaviors at a moderate level. It was noticeable that the mean scores in knowledge about anemia increased at higher education levels. This indicated that school-age children with higher academic grades and those who were older had better abilities to learn, access information, plan, and seek solutions to self-care by themselves [36]. A majority of the sample group seemed to lack knowledge about foods and beverages inhibiting the absorption of iron and those enhancing it. This result is consistent with studies that proposed that among young Ghanaian adolescents aged 10 to 14 years, 18.2 percent knew about iron enhancers while 0.7% knew about iron inhibitors [33]. This is consistent with the study by Chiangkhuntod et al., who studied the knowledge and food consumption behaviors of the residents of Phasi Charoen, Bangkok, and reported that most knew about anemia and food consumption behaviors at a moderate level [25]. For school-age children, the most frequent foods consumed were sweetened milk and commercial foods. The study by Phuengphai et al. demonstrated that children aware of their capability could predict their nutritional health behavior [37]. This indicates that if school-age children clearly understand the benefits of food consumption, appropriate food consumption behavior should follow.

The relationships between knowledge about anemia, food consumption behaviors, and hematocrit level in the sample were at a significantly moderate level. This result is consistent with the study on the associated factors among school-age children in Gondar Town public primary schools, northwest Ethiopia, which proposed that anemia among the school-age children was associated with the insufficient intake of iron-rich foods [38]. Likewise, there is also consistency in the food consumption behaviors among young adolescents who are knowledgeable about anemia conditions due to iron deficiency; they tend to orient

themselves toward consuming foods rich in protein and iron to address the cause and symptoms of anemia. The study also suggested that the correlation of knowledge and food consumption behaviors with anemia is negative, leading to an increase in the capacity for anemia prevention [33]. Emphasizing the significance of knowledge about anemia and food consumption behaviors among school-age children can increase hematocrit levels. In children with sufficient understanding and commitment to proper food consumption behaviors, anemia can be decreased, which is consistent with the suggestions made by Abu-baker et al. and Gebreyesus et al. [39,40]. Therefore, there is a pivotal need to promote understanding and practice among school-age children to establish appropriate food consumption behaviors that can effectively increase the hematocrit percentages of children and decrease anemia cases. Recent studies have recommended iron supplementation and appropriate nutrition management as the health policy was the primary method of treating anemia for a school-aged child in a developing country.

## 5. Conclusions

Anemia remains a persistent public health issue among school-age children. This study found that sufficient knowledge and appropriate food consumption behavior in school-age children at a significantly average level was associated with their hematocrit level. Most of them seemed to have knowledge and food consumption behaviors of a moderate level. As a result, children should have adequate knowledge of this issue and develop food consumption behaviors that can decrease long-term suffering from anemia.

**Author Contributions:** K.K. and P.K. designed the methodology, executed the study in this research. K.K., P.K., O.B., and S.S. assisted with the data collection and analyses. K.K., P.K. wrote the original draft in preparation of the manuscript. K.K. assisted with designing the conceptualization of the study, executing the study, supervising, and writing and editing the final manuscript. K.K. and O.B. assisted with the data analyses and supervised the writing of the manuscript. All authors have read and agreed to the published version of the manuscript.

**Funding:** The authors disclosed receipt of the financial support for the research authorship by the Research Institute for Health Science, Walailak University, Nakhon Si Thammarat, Thailand, with Grant Number WU-IRG-63-012.

**Institutional Review Board Statement:** All procedures performed in studies involving human participants followed the ethical standards of the Ethical Institutional Consideration. The researchers conducted the study following the Declaration of Helsinki. This study received approval from the Ethics Committee on Human Research at Walailak University on 30 April 2020. (WUEC-20-096-01) as required by the process before data collection.

**Informed Consent Statement:** Informed consent was obtained from all subjects involved in the study.

**Data Availability Statement:** Data are available on request from the authors.

**Acknowledgments:** The authors wish to thank the Research Institute for Health Sciences and the Excellence Center of Nursing Institute, Walailak University, for providing invaluable support. We want to express our gratitude to school directors, teachers, parents, and target group participants.

**Conflicts of Interest:** The authors have no conflict of interest.

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
