# Peer review of "Association between Knowledge about Anemia, Food Consumption Behaviors, and Hematocrit Level among School-Age Children in Nakhon Si Thammarat Province, Thailand"

_sustainability, doi:10.3390/su142114599_

Round 1

Reviewer 1 Report

I leave these comments for the authors of this manuscript.

At the end of the introduction, the authors need to indicate the public health relevance of their manuscript. 

The authors need to indicate how they got their sample – a sample size calculation is needed.

How many students did the authors pick in each grade?

The sampling procedure is not clear.

How many schools did you pick in each district, and how many students are in each school?

Measures:

Line120-121: both children and parents consented – I think this varies by the child’s age. – be more specific?

Research instrument

Which instrument(s) did you use in measuring haemoglobin, weight and height – the authors should indicate this?

  • Is not clear how the authors did their composite score for knowledge assessment of anaemia and its prevention and assessment of food consumption behaviour in Children and rarely engaged in that behaviour.

Statistical Section:

It is very elementary for this type of study and journal, and the authors should conduct high-level analysis such as logistic regression or linear regression. The authors should consult a statistician for this. And update all their tables which were poorly presented.

Author Response

Dear Reviewer of the Journal

Please find the attached file that contains our manuscript entitled "Association between Knowledge about Anemia, Food Consumption Behaviors, and Hematocrit Level among School-Age Children in Nakhon Si Thammarat Province, Thailand"  First of all, I am very gladful and thank you very much for the opinions and suggestions of experts to make my manuscript clearer, more appropriate and more accurate. And I have revised already the issues as per the expert's suggestion, and edit of the English language with the proof of experts who use English as their native language.

Response to Reviewer 1 Comments

Point 1: Introduction: At the end of the introduction, the authors need to indicate the public health relevance of their manuscript. 

Response 1: I have additional adjustments already

Point 2: The authors need to indicate how they got their sample – a sample size -students in each grade

Response 2: I have additional adjustments already.

Point 3: Research instrument and Statistical Section

Response 3: I have additional adjustments already

Thank you so much for being so attentive to our manuscript.

Sincerely,

Kiatkamjorn Kusol

*Corresponding author:

Asso. Prof. Dr. Kiatkamjorn Kusol, SFHEA

School of Nursing

Walailak University

Thasala, Nakhon Si Thammarat

Thailand, 80160

Tel: +66 75 672101

E-mail address: kkiatgum@gmail.com

Reviewer 2 Report

This is a descriptive study with 408 students. The survey instruments included the children's demographics and their knowledge of anemia, food consumption behaviors, and hematocrit assessment. It is worth discussing the inclusion or exclusion of girls who may have started menstrual flow. It is worth including current articles discussing iron replacement in childhood.comparison with various world public policies, iron replacement programs, role of primary care should be further explored

Author Response

Dear Reviewer of the Journal

Please find the attached file that contains our manuscript entitled "Association between Knowledge about Anemia, Food Consumption Behaviors, and Hematocrit Level among School-Age Children in Nakhon Si Thammarat Province, Thailand"  First of all, I am very gladful and thank you very much for the opinions and suggestions of experts to make my manuscript clearer, more appropriate and more accurate. And I have revised already the issues as per the expert's suggestion, and edit of the English language with the proof of experts who use English as their native language.

Response to Reviewer 2 Comments

Point 1: It is worth discussing the inclusion or exclusion of girls who may have started menstrual flow.

Response 1: I have additional adjustments already in the inclusion criteria

Point 2: In comparison with various world public policies, and iron replacement programs, the role of primary care should be further explored

Response 2: I have additional adjustments already.

Thank you so much for being so attentive to our manuscript.

Sincerely,

Kiatkamjorn Kusol

*Corresponding author:

Asso. Prof. Dr. Kiatkamjorn Kusol, SFHEA

School of Nursing

Walailak University

Thasala, Nakhon Si Thammarat

Thailand, 80160

Tel: +66 75 672101

E-mail address: kkiatgum@gmail.com

Reviewer 3 Report

The authors have ivestigated the association between knowledge about anemia, food consumption behaviors, and hematocrit Level among School-Age Children in Nakhon Si Thammarat Province, Thailand. The manuscript provides some interesting data and provides the basis for further studies and for recommending public health action for 9-12 year olds that would improve quality of life.

Major Comments/Corrections:

1) Abstract: 'The children knew about anemia, where the mean score was in the middle (Mean= 6.63; SD= 2.51). The mean score on food consumption behaviors was in the middle (Mean= 17.49; SD= 3.68),. Please explain what 'middle' means in the sentence it is not easy to understand.

2) Section 2.3. Please inlcude how the the hematocrit was measured. Was this from a blood count (HCT) or from a packed cell volume (PCV)?

3) Table 1; The units for hct are given as whole numbers (range 25 -47) whereas on line 206 the hct units are given in their conventional format of two decimal place numbers (eg 0.45).

4) Examples of the questionnaires need to be included as supplementary information.

5) It would be useful for the authors to suggest follow-up studies that would demonstrate whether improvements in food, nutritional and anemia knowledge are effective in reducing the incidence of anemia in both a new cohort of children and those in the 12-16 year old age group.

Author Response

Dear Reviewer of the Journal

Please find the attached file that contains our manuscript entitled "Association between Knowledge about Anemia, Food Consumption Behaviors, and Hematocrit Level among School-Age Children in Nakhon Si Thammarat Province, Thailand"  First of all, I am very gladful and thank you very much for the opinions and suggestions of experts to make my manuscript clearer, more appropriate and more accurate. And I have revised already the issues as per the expert's suggestion, and edit of the English language with the proof of experts who use English as their native language.

Response to Reviewer 3 Comments

Point 1: Abstract: The mean score on food consumption behaviors was in the middle.

Response 1: I have additional adjustments

Point 2: Section 2.3. Please include how the hematocrit was measured. Was this from a blood count (HCT) or a packed cell volume (PCV)?

Response 2: The hematocrit level from a packed red cell volume (PCV) and I have additional adjustments already.

Point 3: Table 1; The units for Hct are given as whole numbers (range 25 -47) whereas on line 206 the Hct units are given in their conventional format of two decimal place numbers.

Response 3: I have additional adjustments already.

Point 4: It would be useful for the authors to suggest follow-up studies that would demonstrate whether improvements in food, nutritional, and anemia knowledge are effective in reducing the incidence of anemia in children.

Response 4: Thank you for the suggestion and I plan to follow up on the study the next time.

Thank you so much for being so attentive to our manuscript.

Sincerely,

Kiatkamjorn Kusol

*Corresponding author:

Asso. Prof. Dr. Kiatkamjorn Kusol, SFHEA

School of Nursing

Walailak University

Thasala, Nakhon Si Thammarat

Thailand, 80160

Tel: +66 75 672101

E-mail address: kkiatgum@gmail.com
